# Clinical features of chronic cluster headache based on the third edition of the International Classification of Headache Disorders: A prospective multicentre study

**Soo-Jin Cho[1], Mi Ji Lee[2], Byung-Kun Kim[3], Heui-Soo Moon[4], Pil-Wook Chung[4], Jong-Hee Sohn[5], Soo-Kyoung Kim[6], Yunju Choi[7], Tae-Jin Song[8], Jae-Moon Kim[9], Daeyoung Kim[9], Jeong Wook Park[10], Kwang-Yeol Park[11], Jae-Myun Chung[12], Jin-Young Ahn[13], Byung-Su Kim[14], Kyungmi Oh[15], Dae-Woong Bae[16], Min Kyung Chu[17]*, Chin-Sang Chung[2]**

1 Department of Neurology, Dongtan Sacred Heart Hospital, Hallym University College of Medicine, Hwaseong, Korea, 2 Department of Neurology, Neuroscience Center, Samsung Medical Center, Sungkyunkwan University School of Medicine, Seoul, Korea, 3 Department of Neurology, Eulji University School of Medicine, Seoul, Korea, 4 Department of Neurology, Kangbuk Samsung Hospital, Sungkyunkwan University School of Medicine, Seoul, Korea, 5 Department of Neurology, Chuncheon Sacred Heart Hospital, Hallym University College of Medicine, Chuncheon, Korea, 6 Department of Neurology, Gyeongsang National University College of Medicine, Jinju, Korea, 7 Department of Neurology, Presbyterian Medical Center, Jeonju, Korea, 8 Department of Neurology, Ewha Womans University School of Medicine, Seoul, Korea, 9 Department of Neurology, Chungnam National University College of Medicine, Daejeon, Korea, 10 Department of Neurology, Uijeongbu St.Mary's Hospital, The Catholic University of Korea College of Medicine, Uijeongbu, Korea, 11 Department of Neurology, Chung-Ang University Hospital, Chung-Ang University College of Medicine, Seoul, Korea, 12 Department of Neurology, Seoul Paik Hospital, Inje University College of Medicine, Seoul, Korea, 13 Department of Neurology, Seoul Medical Center, Seoul, Korea, 14 Department of Neurology, Bundang Jesaeng General Hospital, Seongnam, Korea, 15 Department of Neurology, Korea University College of Medicine, Seoul, Korea, 16 Department of Neurology, College of Medicine, The Catholic University of Korea, Suwon, Korea, 17 Department of Neurology, Severance Hospital, Yonsei University School of Medicine, Seoul, Korea

☯ These authors contributed equally to this work.
* chumk@yonsei.ac.kr

**Data Availability Statement:** Public sharing of the data set used in this study is restricted by an IRB restrictions of participating hospital (the

## Abstract

The criterion for the remission period of chronic cluster headache (CCH) was recently revised from < 1 month to < 3 months in the third edition of the International Classification of Headache Disorders (ICHD-3). However, information on the clinical features of CCH based on the ICHD-3 criteria is currently limited. The present study aimed to investigate the clinical features of CCH based on ICHD-3 using data from the Korean Cluster Headache Registry (KCHR). The KCHR is a multicentre prospective registry of patients with cluster headache (CH) from 15 hospitals. Among the 250 participants with CH, 12 and 176 participants were classified as having CCH and episodic cluster headache (ECH), respectively. Among 12 participants with CCH, 6 (50%) had remission periods of < 1 month, and the remaining 6 (50%) had a remission period of 1–3 months. Six participants had CCH from the time of onset of CH, and in the other 6 participants, CCH evolved from ECH. CCH subjects had later age of onset of CH, developed the condition after a longer interval after CH onset, and had more migraine and less nasal congestion and/or rhinorrhoea than ECH subjects.

Institutional Review Board of Samsung Medical Center). To request the data, readers should contact the Korean Headache Society at http://www.headache.or.kr or Dr. Min Kyung Chu, the corresponding author, at chumk@yonsei.ac.kr and Dr. Soo-Jin Cho, the Principal Investigator of KCHR, at dowonc@naver.com.

**Funding:** The authors received no specific funding for this work.

**Competing interests:** SJ Cho was involved as a site investigator of a multicentre trial sponsored by Otsuka Korea, Eli Lilly and Company, and Novartis, worked as an advisory member for Teva, and received research support from Hallym University Research Fund 2016 and a grant from the Korean Neurological Association (KNA-16-MI-09). MK Chu was a site investigator for a multi-centre trial sponsored by Otsuka Korea, Novartis International AG, and Eli Lilly and Company. He worked as an advisory member for Teva, and received lecture honoraria from Allergan Korea, Handok-Teva, and Yuyu Pharmaceutical Company in the past 24 months. The other authors, except for SJ Cho and MK Chu, declare no potential conflicts of interest. This does not alter our adherence to PLOS ONE policies on sharing data and materials.

Clinical features of CCH with remission periods < 1 month were not significantly different from those of CCH with remission periods of 1–3 months, except for the total number of bouts. More current smoking and less diurnal rhythmicity were observed in participants with CCH evolved from ECH compared to those with ECH. In conclusion, the number of subjects with CCH doubled when the revised ICHD-3 criteria were used. Most of clinical characteristics of CCH did not differ when the previous and current version of ICHD was applied and compared. Some clinical features of CCH were different from those of ECH, and smoking may have a role in CH chronification.

## Introduction

Chronic cluster headache (CCH) and episodic cluster headache (ECH) are subtypes of cluster headache (CH) [1, 2]. CCH is defined as CH attacks that typically occur for one year or longer without remission, or with remission periods lasting less than three months. Owing to the severe pain and accompanying symptoms with long-lasting attacks without remission or short remission periods, individuals with CCH experience a high degree of impairment [3]. The criterion for the remission period of CCH was recently revised in the third edition of the International Classification of Headache Disorders (ICHD-3) from < 1 month to < 3 months [2]. However, the clinical features of CCH according to ICHD-3 have only been briefly reported [4]. CCH can begin as CCH from the onset (primary CCH), or can evolve from ECH (secondary CH) [5].

The aim of the present study was to investigate the clinical features of CCH based on the ICHD-3. In addition, we examined the clinical features of CCH according to the remission periods (remission period < 1 month vs. remission period of 1–3 months) and onset patterns (primary vs. secondary).

## Materials and methods

### Study design and participants

This study used data from the Korean Cluster Headache Registry (KCHR). The KCHR is a multicentre registry that used prospective data from consecutively enrolled patients with CH and probable cluster headache (PCH) from 13 university hospitals (8 tertiary referral hospitals and 5 secondary referral hospitals) and 2 secondary referral general hospitals from September 2016 to December 2018. Features of ECH, CCH, and PCH were assessed based on the ICHD-3. The following were the inclusion criteria for the study: a) patients with CH or PCH; b) age ≥ 19 years, and c) full understanding of and agreement to the study protocol. Exclusion criteria were as follows: a) Inability to understand the study protocol, b) current enrolment in other studies, and c) cognitive or psychological difficulty, as per the investigator's judgement, to complete the study. A detailed description of the KCHR study process has been reported in earlier studies [4, 6].

This study protocol was approved by each participating hospital: Dongtan Sacred Heart Hospital (2016-396-I), Samsung Medical Center (2016-09-123), Uijeongbu St.Mary's Hospital (XC16OIMI0087U), Gyeongsang National University College of Medicine (2016-10-022-001), Korea University College of Medicine (KUGH16315), Bundang Jesaeng General Hospital (NR16-04), Seoul Medical Center (2016–013), Kangbuk Samsung Hospital (KBSMC2016-11-032), Eulji University School of Medicine (2016-11-004), Ewha Womans University School of

Medicine (2016-09-021), Seoul Paik Hospital (PAIK 2016-11-003), Presbyterian Medical Center (2016-10-046), Chung-Ang University Hospital (1621-002-264), Chungnam National University College of Medicine (CNUH 2017-12-046), Chuncheon Sacred Heart Hospital (2016–116), Severance Hospital (4-2018-0511), and the Catholic University of Korea (2018-3145-0001).

Written consent forms were obtained from all participating patients. The study was conducted according to the ethical principles of the Declaration of Helsinki. Written informed consent was obtained from all participants.

## Case definition of CH

The diagnosis of CH was based on criteria A to E of the ICHD-3 code 3.1: (A) At least five attacks fulfilling criteria B–D; (B) severe or very severe unilateral orbital, supraorbital, and/or temporal pain lasting 15–180 minutes when untreated; (C) either or both of the following: 1. at least one of the following symptoms or signs ipsilateral to the headache: conjunctival injection and/or lacrimation, nasal congestion and/or rhinorrhoea, eyelid oedema, forehead and facial sweating, forehead and facial flushing, sensation of fullness in the ear, and miosis and/or ptosis, or 2. a sense of restlessness or agitation; (D) attacks with a frequency of between one every other day and eight per day for more than half of the time when the disorder is active; and (E) not better accounted for by another ICHD-3 diagnosis.

If a participant's headache fulfilled criteria A–E of CH and had been occurring for more than 1 year without remission, or with remission periods lasting less than 3 months, a diagnosis of CCH (ICHD-3 code 3.1.2) was made. If a headache fulfilled the criteria of CH, but did not meet the criteria of CCH, the diagnosis of ECH (ICHD-3 code 3.1.1) was assigned. If the first episode of a participant's CH persisted, and could not be diagnosed as CCH or ECH, they were not classified into either ECH or CCH, and a diagnosis of CH (ICHD-3 code 3.1) was made.

If a participant's headache fulfilled all but one of criteria A–D for CH, did not fulfil the ICHD-3 criteria for any other headache disorder, and was not better accounted for by another ICHD-3 diagnosis, he/she was classified as having PCH (code 3.5.1).

## Case definition of migraine

Migraine was diagnosed according to the A–E criteria of migraine without aura (code 1.1) in ICHD-3. If a participant had migraine with aura (code 1.2), she/he was classified as having migraine in the present study.

## Clinical information and CH questionnaire

We assessed each patient's demographic characteristics and clinical information regarding current and previous bouts of CH. Clinical information regarding the current headache episode including location, severity, duration, and frequency of pain, accompanying symptoms, and duration of the cluster period, was evaluated. Previous history of CH including number of years since the onset of CH, age of onset of CH, and the pattern of occurrence were also assessed. Diurnal rhythmicity was assessed based on the participants' report.

Each patient completed a self-administered Headache Impact Test-6 (HIT-6) questionnaire to measure the impact of the headache, the Patient Health Questionnaire-9 (PHQ-9) to assess depression, the Generalized Anxiety Disorder-7 (GAD-7) to assess anxiety, the 3-level version of EQ-5D (EQ-5D-3L) to measure the health-related quality of life, and the Short Form Perceived Stress Scale-4 (PSS-4) to assess psychological stress [7–14].

## Statistical analysis

For continuous variables, the distribution of normality was evaluated by the Kolmogorov-Smirnov test. After normality was confirmed, student's t-tests were used to compare the continuous variables. When normality was not confirmed, continuous variables were analysed by the Mann–Whitney *U* test. For comparing categorical variables, we used Chi-square test. When more than 20% of cells had expected frequencies < 5, Fisher's exact test was used. Statistical significance was set at $p < 0.05$. No statistical power calculation was conducted prior to the study, as the sample size was based on the available data. The Statistical Package for Social Sciences version 23.0 (SPSS 23.0; IBM, Armonk, NY, USA) was used for all statistical analyses.

## Results

### Participants

A total of 250 subjects with CH or PCH were finally enrolled during the study period. Among them, 176 (70.4%), 12 (4.8%), and 27 (10.8%) participants were diagnosed as having ECH (code 3.1.1), CCH (code 3.1.2), and PCH (code 3.5.1), respectively. The remaining 35 (14.0%) participants were classified as having CH without a definite remission period (code 3.1) (Fig 1). Thus, the frequency of CCH among 188 participants with CH with a definitive remission period was 4.6%. Among the 12 participants with CCH, 6 (50.0%) had a remission period of less than 1 month and 6 (50.0%) had a remission period between 1 month and 3 months. Six (50.0%) participants had CCH from the time of onset of CH (primary CCH), and in the remaining six (50.0%) participants, CCH had evolved from ECH (secondary CCH). Out of 12 CCH and 176 ECH participants, 11 (91.7%) and 154 (87.5%) participants, respectively, were enrolled during the active cluster period.

Twenty-seven participants were classified as having PCH and 3 (11.1%) of them reported that the PCH attacks lasted more than one year without a remission period or with a remission period < 3 months.

### Clinical features of CCH

Clinical features of CCH are summarised in Table 1. The median (interquartile range [IQR]) age of participants with CCH was 41.5 (34.0–53.8) years. Of the 12 participants with CCH, 2 (16.7%) were women. The median (IQR) headache frequency per day, visual analogue scale (VAS) score for headache intensity, attack duration (in minutes), cluster period (in weeks), total number of bouts, number of years after CH onset, and age of onset of CH were 2.0 (1.0–2.9), 9.0 (7.6–10.0), 105.0 (60.0–172.5), 4.0 (1.0–6.0), 4.0 (1.0–18.8), 2.5 (1.0–6.8), and 38.0 (28.3–45.5), respectively. The supraorbital area was the most common region in terms of CH pain location (83.3%) followed by the orbital area (75.0%). Conjunctival injection was the most common accompanying symptom (91.7%) followed by restlessness and/or agitation (66.7%). The median (IQR) HIT-6 score with regard to impact of the headache was 71.5 (66.5–74.0) (Table 1). Five participants with CCH had migraine. There was a severe impact of the headache on all participants with CCH (HIT-6 ≥ 60) [14]. The frequency of CCH participants experiencing severe impact of headache was marginally higher compared with that of ECH participants; the difference was marginally insignificant (100.0% vs. 81.3%, $p$ = 0.091).

### Comparison of clinical features of CCH and ECH

We compared the clinical features of CCH with those of ECH. The headache frequency per day, VAS score for headache, duration of CH attack (in minutes), and total number of bouts

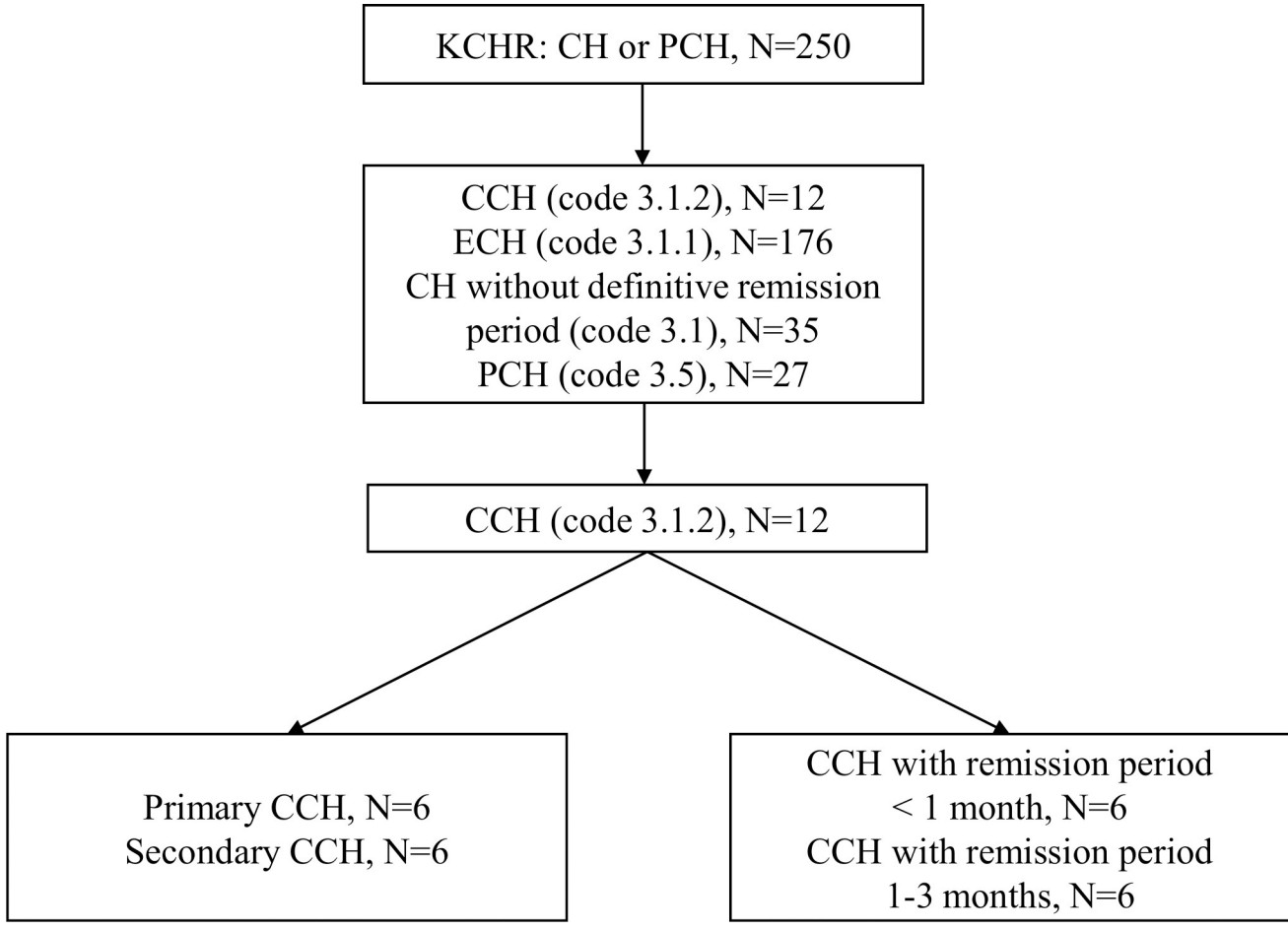

**Fig 1. Flow chart of study participants.** KCHR: Korean Cluster Headache Registry, CH: cluster headache; CCH: chronic cluster headache; ECH: episodic cluster headache; PCH: probable cluster headache.

were not significantly different between CCH and ECH subjects. Nevertheless, the age of onset of CH (38.0 [28.3–45.5] vs. 24.0 [18.0–32.8] years, $p = 0.003$) was significantly higher and number of years after CH onset (2.5 [1.0–6.75] vs. 10.0 [5.0–16.0], $p<0.001$) was significantly lower in participants with CCH compared to participants with ECH. More number of patients with CCH had migraine than those with ECH (41.7% vs. 10.8%, $p = 0.019$). Distribution of location of pain and frequency of accompanying symptoms were not significantly different between CCH and ECH patients, except for nasal congestion and/or rhinorrhoea. Nasal congestion and/or rhinorrhoea was less prevalent in CCH subjects compared with ECH subjects (25.0% vs. 64.0%, $p = 0.012$). The impact of headache (HIT-6), perceived stress (PSS), quality of life (EQ-5D-3L), depression (PHQ-9), and anxiety (GAD-7) were not significantly different between CCH and ECH subjects (Table 1).

## Comparison of clinical features of CCH with remission period of < 1 month and CCH with remission period of 1–3 months

The total number of bouts was significantly lower in CCH with a remission period of < 1 month than in CCH with a remission period of 1–3 months (6.0 [5.0–9.5] vs. 17.5 [1.8–57.0],

**Table 1. Clinical features of participants with chronic cluster headache and episodic cluster headache.**

| | CCH, n = 12 | ECH, n = 176 | *p*-value |
|---|---|---|---|
| Age (years) | 41.5 (34.0–53.8)* | 39.0 (32.0–45.0)* | 0.345 |
| Women, n (%) | 2 (16.7) | 24 (14.0) | 0.679 |
| Body Mass Index | 24.3 (22.9–26.9)* | 23.8 (22.0–25.7)* | 0.382 |
| Attack frequency per day | 2.0 (1.0–2.9)* | 1.1 (1.0–3.0)* | 0.404 |
| VAS, median | 9.0 (7.6–10.0)* | 9.0 (8.0–10.0)* | 0.308 |
| Attack duration (minutes) | 105.0 (60.0–172.5) * | 60.0 (60.0–120.0)* | 0.147 |
| Cluster period (weeks) | 4.0 (1.0–6.0)* | 4.00 (3.0–7.50)* | 0.560 |
| Total number of bouts | 4.0 (1.0–18.8)* | 7.00 (3.0–12.0)* | 0.306 |
| Years after CH onset | 2.5 (1.0–6.8)* | 10.0 (5.0–16.0)* | <0.001 |
| Age of onset of CH | 38.0 (28.3–45.5)* | 24.0 (18.0–32.8)* | 0.003 |
| Change of pain side during a single bout period, n (%) | 1 (8.3) | 9 (5.2) | 0.983 |
| Currently smoking, n (%) | 4 (33.3%) | 79 (45.9) | 0.397 |
| Diurnal rhythmicity, n (%) | 3 (25.0) | 90 (53.3) | 0.140 |
| Migraine, n (%) | 5 (41.7) | 19 (10.8) | 0.019 |
| Location, n (%) | | | |
| Orbital | 9 (75.0) | 143 (83.1) | 0.472 |
| Temporal | 8 (66.7) | 89 (51.7) | 0.317 |
| Supraorbital | 10 (83.3) | 97 (56.4) | 0.077 |
| Accompanying symptoms, n (%) | | | |
| Conjunctival injection and/or lacrimation | 11 (91.7) | 152 (88.4) | 1.000 |
| Nasal congestion and/or rhinorrhoea | 3 (25.0) | 110 (64.0) | 0.012 |
| Eyelid oedema | 3 (25.0) | 50 (29.1) | 1.000 |
| Forehead and facial sweating | 5 (14.7) | 54 (31.4) | 0.461 |
| Miosis and ptosis | 0 (0.0) | 6 (3.5) | 1.000 |
| Restlessness/ agitation | 2 (16.7) | 40 (23.3) | 0.737 |
| Headache Impact Test-6 | 71.5 (66.5–74.0)* | 69.0 (64.0–76.0)* | 0.547 |
| Severe impact of headache (Headache Impact Test-6 score ≥ 60) | 12 (100.0) | 143 (81.3) | 0.091 |
| Perceived Stress Scale | 8.0 (6.3–8.8)* | 7.0 (4.0–8.0)* | 0.149 |
| EQ-5D-3L | 0.91 (0.68–0.98)* | 0.91 (0.82–1.00)* | 0.591 |
| Patient Health Questionnaire-9 | 8.0 (1.8–8.8)* | 5.0 (3.0–11.0)* | 0.826 |
| Generalized Anxiety Disease-7 | 6.5 (1.0–9.8)* | 7.0 (3.0–12.0)* | 0.265 |

CCH, chronic cluster headache; ECH, episodic cluster headache; VAS, visual analogue scale; EQ-5D-3L, The 3-level version of European quality of life-5 dimensions

*: median (25%–75% interquartile range)

*p* = 0.046). Age, frequency of headaches per day, VAS score for headache intensity, duration of CH attack (in minutes) and total number of bouts, frequency of smoking, distribution of pain location, accompanying symptoms, and impact of headache and perceived stress were not significantly different between CCH with a remission period < 1 month and CCH with a remission period of 1–3 months (Table 2).

## Comparison of clinical features of primary CCH and secondary CCH

We compared the clinical features of CCH according to the onset pattern (primary CCH vs. secondary CCH). The total number of bouts was significantly higher in secondary CCH than

**Table 2. Clinical features of participants with chronic cluster headache with a remission period < 1 month and chronic cluster headache with a remission period of 1–3 months.**

| | CCH with a remission period < 1 month, n = 6 | CCH with a remission period of 1–3 month, n = 6 | *p*-value |
|---|---|---|---|
| Age (years) | 40.0 (35.3–53.0)* | 41.5 (30.0–58.0) * | 0.873 |
| Women, n (%) | 0 (0.0) | 2 (33.3) | 0.454 |
| Body Mass Index | 25.9 (22.3–27.3)* | 23.6 (22.6–26.3)* | 0.522 |
| Attack frequency per day | 2.3 (1.8–3.7)* | 1.5 (0.5–4.0)* | 0.417 |
| VAS, median | 9.5 (8.5–10.0)* | 8.5 (7.4–9.3)* | 0.310 |
| Attack duration (minutes) | 90.0 (52.5–157.5)* | 105.0 (82.5–180.0)* | 0.463 |
| Cluster period (weeks) | 5.0 (1.0–5.0)* | 5.5 (3.3–16.5)* | 0.175 |
| Total number of bouts | 6.0 (5.0–9.50)* | 17.5 (1.8–57.0)* | 0.046 |
| Years after CH onset | 1.5 (1.0–7.3)* | 4.0 (1.8–6.5)* | 0.462 |
| Age of onset of CH | 44.0 (12.7–53.0)* | 38.0 (24.0–49.8)* | 0.810 |
| Change of pain side during a single bout period, n (%) | 0 (0.0) | 1 (16.7) | 1.000 |
| Currently smoking, n (%) | 0 (0.0) | 1 (16.7) | 1.000 |
| Diurnal rhythmicity, n (%) | 3 (50.0) | 3 (50.0) | 0.545 |
| Migraine, n (%) | 3 (50.0) | 2 (33.3) | 1.000 |
| Location, n (%) | | | |
| Orbital | 1 (16.7) | 2 (33.3) | 1.000 |
| Supraorbital | 4 (66.7) | 6 (100.0) | 0.455 |
| Temporal | 2 (33.3) | 4 (66.7) | 1.000 |
| Accompanying symptoms, n (%) | | | |
| Conjunctival injection and/or lacrimation | 5 (83.3) | 6 (100.0) | 1.000 |
| Nasal congestion and/or rhinorrhoea | 1 (16.7) | 2 (33.3) | 1.000 |
| Eyelid oedema | 2 (33.3) | 1 (16.7) | 1.000 |
| Forehead and facial sweating | 2 (33.3) | 3 (50.0) | 1.000 |
| Miosis and ptosis | 0 (0.0) | 0 (0.0) | 1.000 |
| Restlessness/ agitation | 1 (16.7) | 1 (16.7) | 1.000 |
| Headache Impact Test-6 | 71.5 (67.5–74.5)* | 71.5 (64.8–75.0)* | 0.809 |
| Severe impact of headache (Headache Impact Test-6 score ≥ 60) | 6 (100.0) | 6 (100.0) | 1.000 |
| Perceived Stress Scale | 5.0 (5.8–9.5)* | 7.5 (5.8–8.3)* | 0.567 |
| EQ-5D-3L | 0.84 (0.63–1.00)* | 0.91 (0.75–0.93)* | 0.745 |
| Patient Health Questionnaire-9 | 7.5 (0.8–12.3)* | 8.0 (3.0–8.8)* | 1.000 |
| Generalized Anxiety Disease-7 | 3.5 (0.8–12.0)* | 8.5 (0.8–9.3)* | 1.000 |

CCH, chronic cluster headache; VAS, visual analogue scale; EQ-5D-3L, The 3-level version of European quality of life-5 dimensions

*: median (25%–75% interquartile range)

in primary CCH (8.0 [1.0–8.0] vs. 16.5 [5.0–57.0], *p* = 0.015). Frequency of diurnal rhythmicity was higher in primary CCH; the difference was marginally insignificant (66.7% vs. 0.0%, *p* = 0.061). The number of current smokers was higher in secondary CCH than in primary CCH, but the difference was marginally insignificant (16.7% vs. 66.7%, *p* = 0.091). Frequency of migraine occurrence was not significantly different between the two groups. Age, headache frequency per day, VAS score for headache intensity, attack duration (in minutes), cluster period (in weeks), distribution of pain location, accompanying symptoms, and impact of headache, perceived stress, depression, and anxiety were not significantly different between primary and secondary CCH (Table 3).

**Table 3. Clinical features of participants with primary chronic cluster headache and secondary chronic cluster headache.**

| | Primary CCH, n = 6 | Secondary CCH, n = 6 | *p*-value |
|---|---|---|---|
| Age (years) | 39.5 (35.3–47.5)* | 44.5 (30.0–59.8)* | 0.688 |
| Women, n (%) | 2 (33.0) | 0 (0.0) | 0.455 |
| Body Mass Index | 24.3 (22.9–26.9)* | 24.3 (22.0–28.2)* | 1.000 |
| Attack frequency per day | 2.0 (1.0–2.1)* | 2.8 (0.5–7.0)* | 0.372 |
| VAS, median | 8.5 (7.0–10.0)* | 9.0 (8.6–10.0)* | 0.454 |
| Attack duration (minutes) | 120.0 (82.5–157.5)* | 75.0 (52.5–180.0)* | 0.596 |
| Cluster period (weeks) | 52.0 (36.5–172.0)* | 8.0 (1.8–47.3)* | 0.865 |
| Total number of bouts | 8.0 (1.0–8.0)* | 16.5 (5.0–57.0)* | 0.015 |
| Years after CH onset | 1.5 (1.0–3.5)* | 5.5 (2.5–7.3)* | 0.141 |
| Age of onset of CH | 36.5 (28.8–46.5)* | 40.5 (24.0–50.0)* | 0.873 |
| Change of pain side during a single bout period, n (%) | 0 (0.0) | 1 (16.7) | 1.000 |
| Currently smoking, n (%) | 1 (16.7) | 4 (66.7) | 0.091 |
| Diurnal rhythmicity, n (%) | 4 (66.7) | 0 (0.0) | 0.061 |
| Migraine, n (%) | 4 (66.7) | 1 (16.7) | 0.242 |
| Location, n (%) | | | |
| Orbital | 3 (50.0) | 6 (100.0) | 0.182 |
| Temporal | 5 (83.3) | 5 (83.3) | 1.000 |
| Supraorbital | 3 (50.0) | 5 (83.0) | 0.545 |
| Accompanying symptoms, n (%) | | | |
| Conjunctival injection and/or lacrimation | 5 (83.3) | 6 (100.0) | 1.000 |
| Nasal congestion and/or rhinorrhoea | 2 (33.3) | 1 (16.7) | 1.000 |
| Eyelid oedema | 0 (0.0) | 3 (50.0) | 0.182 |
| Forehead and facial sweating | 3 (50.0) | 2 (33.3) | 1.000 |
| Miosis and ptosis | 0 (0.0) | 0 (0.0) | 1.000 |
| Restlessness/ agitation | 1 (16.7) | 1 (16.7) | 1.000 |
| Headache Impact Test-6 | 71.5 (67.0–72.5)* | 72.5 (65.8–76.5)* | 0.520 |
| Severe impact of headache (Headache Impact Test-6 score ≥ 60) | 6 (100.0) | 6 (100.0) | 1.000 |
| Perceived Stress Scale | 8.50 (6.5–9.5)* | 7.5 (0.8–9.0)* | 0.191 |
| EQ-5D-3L | 0.89 (0.63–0.93)* | 0.91 (0.68–1.00)* | 0.516 |
| Patient Health Questionnaire-9 | 8.0 (6.3–12.3)* | 4.50 (0.0–8.8)* | 0.253 |
| Generalized Anxiety Disease-7 | 7.5 (0.8–12.0)* | 5.0 (0.8–9.0)* | 0.420 |

CCH, chronic cluster headache; ECH, episodic cluster headache; VAS, visual analogue scale; EQ-5D-3L, The 3-level version of European quality of life-5 dimensions

*: median (25%–75% interquartile range)

## Comparison of clinical features of secondary CCH and ECH

Secondary CCH had a greater association with current smoking (66.7% vs. 45.9%, *p* = 0.025), and had less diurnal rhythmicity (0.0% vs. 51.1%, *p* = 0.011) than ECH. The number of years after onset of CH was lesser in secondary CCH than in secondary ECH (5.5 [2.5–7.3] vs. 10.0 [5.0–16.0], *p* = 0.041). Nasal congestion and/or rhinorrhoea were less prevalent in secondary CCH, but the difference was marginally insignificant (16.7% vs. 64.0%, *p* = 0.068). Other clinical features were not significantly different between secondary CCH and ECH (S1 Table).

## Discussion

The main findings of the present study were as follows: 1) 6.4% of CH participants with a definitive remission period had CCH. The number of CCH participants doubled after revision

of the remission period criterion in ICHD-3; 2) Some clinical features of CCH were different from those of ECH including total number of bouts, number of years after the onset of CH, age of onset of CH, nasal congestion and/or rhinorrhoea during CH attacks, and presence of migraine; 3) Approximately one-tenth of the participants with PCH experienced PCH attacks lasting more than one year with remission period < 3 months.

The diagnostic criteria of CCH were recently revised in ICHD-3. While the criterion for the duration of CH attacks did not change to more than one year, the criterion for the remission period increased from < 1 month to < 3 months. Thus, the proportion of patients with CCH diagnosis may increase with the use of ICHD-3. In the present study, the number of CCH subjects increased from 6 to 12 among all CH participants with a definite remission period. It was also observed that demographics and clinical features of CCH with a remission period of < 1 month were not significantly different from those of CCH with a remission period of 1–3 months, except for the total number of bouts (Table 2). These findings suggest that the diagnostic criteria of CCH in ICHD-3 were properly revised with respect to clinical features.

Although the criteria for clinical features of CCH attacks are the same as those of ECH attacks, differences in the clinical features between the two have been reported. Patients with CCH have reported milder headache intensity, more changes in pain side during a single bout, and less rhinorrhoea than those with ECH [15, 16]. In our study, nasal congestion and/or rhinorrhoea was less prevalent in CCH than in ECH subjects. In addition, we found that migraine was more prevalent in CCH than in ECH subjects (Table 1); a similar trend was observed in a Swedish study [17]. Nevertheless, headache intensity and change in the pain side during a single bout period were not significantly different between CCH and ECH subjects in the present study. The difference in clinical features between the present study and previous studies may be owing to differences in the study population and evaluation methods.

The present study demonstrated that all participants with CCH had a severe functional impact of headache. The prevalence of participants experiencing severe impact of headache was higher in CCH than in ECH, although statistically insignificant. These findings are in agreement with a French study, wherein, it was observed that 74.1% of participants with CCH endured a severe impact of headache [18]. Another German study demonstrated similar findings, and reported that patients with CCH and active ECH were severely impaired [3]. Thus, we could verify that patients with CCH experience a substantial functional impact of headache.

It was observed in previous studies that 45–54% cases of CCH were evolved from ECH (secondary CCH). A longer course of CH (> 20 years), late age of onset, exposure to tobacco, and longer duration of the cluster period (> 8 weeks) were reported to be associated with the transition from ECH to CCH [17, 19–21]. In our study, we observed that in half the CCH participants, the CCH had evolved from ECH. Secondary CCH showed a greater relationship with current smoking, and had less diurnal rhythmicity than primary CCH, although the statistical significance was not reached (Table 2). On comparing secondary CCH to ECH, current smoking was more prevalent and diurnal rhythmicity was less prevalent in secondary CCH (S1 Table). Therefore, current smoking might have an impact on progression to CCH from ECH [21, 22]. Considering that patients with CCH had a high functional impact of headache, identifying predictive factors with regard to transition from ECH and CCH would be an important step in reducing the burden of CCH in addition to the understanding the pathophysiology of CCH [1, 3, 17]. Further studies for identifying factors responsible for transition from ECH to CCH in various conditions would be needed.

Diurnal rhythmicity is an important feature of CH, but the exact mechanisms are unclear. A recent Danish study reported that diurnal rhythmicity was less dominant in CCH than in ECH. The study excluded CH participants with no rhythmicity and did not separately assess

CCH participants according to the onset pattern (primary or secondary) [23]. Considering the fact that approximately half of the CCH cases are secondary, the reduced diurnal rhythmicity in secondary CCH observed in the present study was in agreement with the findings of the Danish study.

The proportion of participants with CCH (6.4%) in the present study was lower than that in the previous studies in Western countries. The proportion of subjects with CCH ranged from 11.4% to 36.6% in studies in European and North American countries [15–18, 24–26]. Nevertheless, the proportion of patients with CCH in our study was similar to that observed in Asian countries. CCH was observed in 3.5% of CH patients in a Japanese cohort based on ICHD-2 criteria [27]. Further, CCH was present in 7.5% of Chinese patients with CH [28]. The similarity in the proportion of CCH cases in our study with that of Asian studies suggests that the present study evaluated CCH properly. To date, the pathophysiology of CH chronification has not been elucidated yet. Considering the great difference in the prevalence of CCH among countries, genetic and environmental factors might play a role in developing CCH, whereas the prevalence of chronic migraine is similar worldwide [29]. In addition, functional connectivity of networks involving pain processing or pain modulation may be a factor which determines chronicity of CH, as evidenced in other pain disorders and chronic migraine [30, 31]. One study using resting-state functional magnetic resonance imaging revealed a possible dysfunction in the pain processing in interictal CH patients [32].

This study has several limitations. Firstly, the study was a cross-sectional study, and could not identify factors for a cause-and-effect relationship. The classification of ECH and CCH was based on patients' description of their clinical courses, which may lead a possibility of recall bias. Nevertheless, comparing clinical features between CCH and ECH and between secondary CCH and ECH provided a clue for identifying factors related to CCH and secondary CCH. Further studies in a longitudinal setting would provide more information regarding factors related to CCH or secondary CCH. Secondly, we did not investigate the frequency of smoking and amount of alcohol consumption. A previous study reported that smoking (> 20 cigarettes) and alcohol consumption (> 100 g) were more prevalent in CCH than in ECH patients [33]. Another study identified that cigarette smoking (> 20 cigarettes) and alcohol intake (50–100 g) were more frequently reported in secondary CCH than in primary CCH patients [5]. However, in our study, we only evaluated the status of smoking and alcohol intake, and did not quantify it. Thirdly, the sample size of the CCH group was not high, and the insignificance observed in some analyses may have been due to the limited sample size. In other words, our findings might have type II error. However, we found several significant findings using non-parametric analyses for continuous variables and Fisher's exact tests for categorical variables, which are valid even with small sample sizes. Finally, the validation of the ICHD-3 criteria was based on the comparison between 6 and 12 patients in our study. This may be affected by biases because the number of patients was small and the two groups were not completely independent to each other.

Our study also has several strengths. Firstly, this study evaluated the demographic and clinical features of CCH according to the ICHD-3 criteria that were recently revised. Information on the clinical features of CCH based on ICHD-3 criteria is currently limited. Secondly, we compared the clinical features of CCH and ECH, and those of secondary CCH and ECH, and found that some clinical features of CCH and secondary CCH were significantly different from those of ECH. These findings may provide clues for identifying significant factors responsible for transition from ECH to CCH. Thirdly, we included participants with PCH, which has rarely been evaluated in previous studies. We observed that approximately 10% of participants with PCH had remission periods < 3 months. Findings in the present study may

provide information on the clinical features of CCH based on ICHD-3 that was recently revised.

## Conclusions

The number of subjects with CCH doubled when the revised ICHD-3 criteria were used. The clinical features of CCH with a remission period of < 1 month and a remission period of 1–3 months, which were previously classified into ECH but newly included into CCH in ICHD-3, were not significantly different, except for the total number of bouts. Some clinical features of CCH were different from those of ECH, including lesser nasal congestion and/or rhinorrhoea, longer interval after onset of CH, later age of onset of CH, and higher incidence of migraine. Current smoking may have a role in the transition from ECH to CCH.

## Supporting information

**S1 Table. Clinical features of participants with secondary chronic cluster headache and episodic chronic cluster headache.**
(DOCX)

## Acknowledgments

The authors would like to express their appreciation to all participants in the study.

## Author Contributions

**Conceptualization:** Soo-Jin Cho, Pil-Wook Chung, Min Kyung Chu.

**Data curation:** Mi Ji Lee, Byung-Kun Kim, Heui-Soo Moon, Pil-Wook Chung, Jong-Hee Sohn, Soo-Kyoung Kim, Yunju Choi, Tae-Jin Song, Jae-Moon Kim, Daeyoung Kim, Jeong Wook Park, Kwang-Yeol Park, Jae-Myun Chung, Jin-Young Ahn, Byung-Su Kim, Kyungmi Oh, Dae-Woong Bae, Min Kyung Chu, Chin-Sang Chung.

**Formal analysis:** Soo-Jin Cho, Min Kyung Chu.

**Funding acquisition:** Soo-Jin Cho.

**Investigation:** Soo-Jin Cho, Mi Ji Lee, Byung-Kun Kim, Heui-Soo Moon, Pil-Wook Chung, Jong-Hee Sohn, Soo-Kyoung Kim, Yunju Choi, Tae-Jin Song, Jae-Moon Kim, Daeyoung Kim, Jeong Wook Park, Kwang-Yeol Park, Jae-Myun Chung, Jin-Young Ahn, Byung-Su Kim, Kyungmi Oh, Dae-Woong Bae, Min Kyung Chu, Chin-Sang Chung.

**Writing – original draft:** Soo-Jin Cho, Mi Ji Lee, Min Kyung Chu.

**Writing – review & editing:** Soo-Jin Cho, Mi Ji Lee, Byung-Kun Kim, Heui-Soo Moon, Pil-Wook Chung, Jong-Hee Sohn, Soo-Kyoung Kim, Yunju Choi, Tae-Jin Song, Jae-Moon Kim, Daeyoung Kim, Jeong Wook Park, Kwang-Yeol Park, Jae-Myun Chung, Jin-Young Ahn, Byung-Su Kim, Kyungmi Oh, Dae-Woong Bae, Min Kyung Chu, Chin-Sang Chung.

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
