## [Decision Letter · Decision Letter 0]

17 Jul 2019

PONE-D-19-17566

Clinical features of chronic cluster headache based on the third edition of the International Classification of Headache Disorders: A prospective multicentre study

PLOS ONE

Dear Dr. Min Kyung Chu,

Thank you for submitting your manuscript to PLOS ONE. After careful consideration, we feel that it has merit but does not fully meet PLOS ONE’s publication criteria as it currently stands. Therefore, we invite you to submit a revised version of the manuscript that addresses the points raised during the review process.

ACADEMIC EDITOR: Our expert reviewer(s) have recommended some minor revisions to your manuscript. Therefore, I invite you to respond to the reviewer(s)' comments as below and revise your manuscript.

We would appreciate receiving your revised manuscript by Aug 31 2019 11:59PM. To enhance the reproducibility of your results, we recommend that if applicable you deposit your laboratory protocols in protocols.io, where a protocol can be assigned its own identifier (DOI) such that it can be cited independently in the future. For instructions see: http://journals.plos.org/plosone/s/submission-guidelines#loc-laboratory-protocols

We look forward to receiving your revised manuscript.

Kind regards,

Wisit Cheungpasitporn, MD, FACP

University of Mississippi Medical Center

Twitter: @wisit661 Email: wcheungpasitporn@gmail.com 

Academic Editor

PLOS ONE

Journal Requirements:

1. Thank you for including your competing interests statement; "SJ Cho was involved as a site investigator of a multicentre trial sponsored by Otsuka Korea, Eli Lilly and Company, and Novartis, worked as an advisory member for Teva, and received research support from Hallym University Research Fund 2016 and a grant from the Korean Neurological Association (KNA-16-MI-09).

MK Chu was a site investigator for a multi-centre trial sponsored by Otsuka Korea, Novartis International AG, and Eli Lilly and Company. He worked as an advisory member for Teva, and received lecture honoraria from Allergan Korea, Handok-Teva, and Yuyu Pharmaceutical Company in the past 24 months.

The other authors, except for SJ Cho and MK Chu, declare no potential conflicts of interest."

Reviewers' comments:

Reviewer's Responses to Questions

**Comments to the Author**

1. Is the manuscript technically sound, and do the data support the conclusions?

Reviewer #1: Yes

Reviewer #2: Yes

Reviewer #3: Yes

2. Has the statistical analysis been performed appropriately and rigorously? 

Reviewer #1: Yes

Reviewer #2: Yes

Reviewer #3: Yes

3. Have the authors made all data underlying the findings in their manuscript fully available?

Reviewer #1: Yes

Reviewer #2: Yes

Reviewer #3: Yes

4. Is the manuscript presented in an intelligible fashion and written in standard English?

Reviewer #1: Yes

Reviewer #2: Yes

Reviewer #3: Yes

5. Review Comments to the Author

Reviewer #1: The definition of chronic cluster headache has been changed in the last ICHD3 classification, increasing the time for relapse period from 1 to 3 months. The authors tested of the Korean cluster headache registry the effect of this modification in order to test the new definition and the clinical featurs of the CCH.

The most relevant observation is that the number of patients with CCH doubles with the new criteria.

The number of patients with CCH vs ECH are much less (176 vs 12) but the authors were able, with the statistical correction to still be able to find some significant diferences between the two populations. In particular they found out that the patients with CCH has higher age and lower CH history vs patients with ECH, they have less nasal congestion and rhinorrhea, and have more comorbidity with migraine. Interestingly the authors did not find any differenced in the population reclassified as CCH with the new criteria, meaning that the patients are probably right classified with the new criteria.

They observed how smoke habit may lead CH to become chronic lathough to clearly significant (probably due to the small numeber of pts).

The limits of the study where well stated.

- It would be interesting in the discussion to speculate if there might be a possible role of chronic pain networks disfunction in both conditions (Colombo B et al Neurol SCI 2015).

Reviewer #2: This is the multi-center, prospective study to investigate the clinical feature of chronic cluster headache using the new definition in hospitala in south korea. Overall, manuscript is well written. I cannot identify the major of this study.

Reviewer #3: -Please discuss in the limitations the types of biases the findings of this study may be influenced by

-The limitations of the study particularly in regards to the validation of diagnosis needs to be more robust.

-Some revision of the English language is needed.

“Diagnosis of ECH (code 3.1.1) and CCH (code 3.1.2) were based on criteria” is not correct in grammar.

“huge” should be avoided in academic writing.

6. PLOS authors have the option to publish the peer review history of their article (what does this mean?). If published, this will include your full peer review and any attached files.

Reviewer #1: No

Reviewer #2: No

Reviewer #3: No

---

## [Author Response · Author response to Decision Letter 0]

25 Jul 2019

Author Response Letter

We thank the reviewers for their constructive comments. We believe our paper has been improved based on the Reviewers’ comments. In this letter, we have provided a point-by-point response to each comment below. Textual changes based on the Reviewers’ comments are tracked in the revised manuscript. We hope our paper can meet the journal’s requirements.

Reviewer #1

1. The definition of chronic cluster headache has been changed in the last ICHD3 classification, increasing the time for relapse period from 1 to 3 months. The authors tested of the Korean cluster headache registry the effect of this modification in order to test the new definition and the clinical features of the CCH.

Response: We thank the reviewer for his/her time and efforts in reviewing our paper. 

2. The most relevant observation is that the number of patients with CCH doubles with the new criteria.

Response: We agree with the reviewer that this is the major finding of our study. The latest ICHD-3 criteria broadened the spectrum of CCH without compromising its functional impact. 

3. The number of patients with CCH vs ECH are much less (176 vs 12) but the authors were able, with the statistical correction to still be able to find some significant diferences between the two populations. In particular they found out that the patients with CCH has higher age and lower CH history vs patients with ECH, they have less nasal congestion and rhinorrhea, and have more comorbidity with migraine. Interestingly the authors did not find any difference in the population reclassified as CCH with the new criteria, meaning that the patients are probably right classified with the new criteria. They observed how smoke habit may lead CH to become chronic lathough to clearly significant (probably due to the small numeber of pts). The limits of the study where well stated.

Response: We appreciate the reviewer for properly summarizing our findings and giving us constructive comments. 

4. It would be interesting in the discussion to speculate if there might be a possible role of chronic pain networks dysfunction in both conditions (Colombo B et al Neurol SCI 2015).

Response: Thank you for the insightful suggestion. We have added discussions regarding the role of pain networks in CH chronification. (Line 307 – 314 in the track-changes version)

Reviewer #2: This is the multi-center, prospective study to investigate the clinical feature of chronic cluster headache using the new definition in hospitala in south korea. Overall, manuscript is well written. I cannot identify the major of this study.

Response: We thank the reviewer for his/her time and efforts in reviewing our paper and constructive comments. We are pleased that the reviewer was satisfied with our paper. 

Reviewer #3: 

1. Please discuss in the limitations the types of biases the findings of this study may be influenced by

Response: We thank the reviewer for his/her time and efforts in reviewing our paper. The classification of ECH and CCH was based on patients’ description of their clinical courses, so recall bias may be present. Our study may also have type II error because the number of patients with CCH was small. We have added these information in the Limitations section. (Line 316 – 318, 328 – 329 in the track-changes version)

2. The limitations of the study particularly in regards to the validation of diagnosis needs to be more robust.

Response: Thank you for this comment. In our study, the validation of the ICHD-3 criteria was based on the comparison between 6 and 12 patients. This may be affected by biases because the number of patients was small, and the two groups were not completely independent to each other. We have added this in the Limitation section. (Line 332 – 334 in the track-changes version)

3. Some revision of the English language is needed.

3-1. “Diagnosis of ECH (code 3.1.1) and CCH (code 3.1.2) were based on criteria” is not correct in grammar.

Response: Thank you for pointing this out. The expressions have been revised. (Line 96 – 97, 108 – 112 in the track-changes version)

3-1. “huge” should be avoided in academic writing.

Response: Thank you for the suggestion. This expression has been also revised. (Line 278 in the track-changes version)

---

## [Decision Letter · Decision Letter 1]

1 Aug 2019

Clinical features of chronic cluster headache based on the third edition of the International Classification of Headache Disorders: A prospective multicentre study

PONE-D-19-17566R1

Dear Dr. Min Kyung Chu,

We are pleased to inform you that your manuscript has been judged scientifically suitable for publication and will be formally accepted for publication once it complies with all outstanding technical requirements.

With kind regards,

Wisit Cheungpasitporn, MD, FACP

University of Mississippi Medical Center

Twitter: @wisit661 Email: wcheungpasitporn@gmail.com 

Academic Editor

PLOS ONE

Additional Editor Comments:

I want to commend the authors on their superb efforts to revise the manuscript according to all reviewers’ suggestions. The quality of the manuscript has improved substantially.

Reviewers' comments:

Reviewer's Responses to Questions

**Comments to the Author**

1. If the authors have adequately addressed your comments raised in a previous round of review and you feel that this manuscript is now acceptable for publication, you may indicate that here to bypass the “Comments to the Author” section, enter your conflict of interest statement in the “Confidential to Editor” section, and submit your "Accept" recommendation.

Reviewer #1: All comments have been addressed

Reviewer #2: (No Response)

Reviewer #3: All comments have been addressed

2. Is the manuscript technically sound, and do the data support the conclusions?

Reviewer #1: Yes

Reviewer #2: (No Response)

Reviewer #3: Yes

3. Has the statistical analysis been performed appropriately and rigorously? 

Reviewer #1: Yes

Reviewer #2: (No Response)

Reviewer #3: Yes

4. Have the authors made all data underlying the findings in their manuscript fully available?

Reviewer #1: Yes

Reviewer #2: (No Response)

Reviewer #3: Yes

5. Is the manuscript presented in an intelligible fashion and written in standard English?

Reviewer #1: Yes

Reviewer #2: (No Response)

Reviewer #3: Yes

6. Review Comments to the Author

Reviewer #1: (No Response)

Reviewer #2: All of my comments have been addressed properly. I have no further recommendation to improve this manuscript.

Reviewer #3: It is an interesting and relevant article. I consider it a useful contribution in its field. Revised manuscript is acceptable for publication.

7. PLOS authors have the option to publish the peer review history of their article (what does this mean?). If published, this will include your full peer review and any attached files.

Reviewer #1: No

Reviewer #2: No

Reviewer #3: No

---

## [Editor Report · Acceptance letter]

19 Aug 2019

PONE-D-19-17566R1 

Clinical features of chronic cluster headache based on the third edition of the International Classification of Headache Disorders: A prospective multicentre study 

Dear Dr. Chu:

I am pleased to inform you that your manuscript has been deemed suitable for publication in PLOS ONE. Congratulations! Your manuscript is now with our production department. 

With kind regards,

on behalf of

Dr. Wisit Cheungpasitporn 

Academic Editor

PLOS ONE